# The Influence of Reduced Graphene Oxide on the Texture and Chemistry of N,S-Doped Porous Carbon. Implications for Electrocatalytic and Energy Storage Applications

**DOI:** 10.3390/nano13162364

**Published:** 2023-08-18

**Authors:** Samantha K. Samaniego Andrade, Shiva Shankar Lakshmi, István Bakos, Szilvia Klébert, Robert Kun, Miklós Mohai, Balázs Nagy, Krisztina László

**Affiliations:** 1Department of Physical Chemistry and Materials Science, Faculty of Chemical Technology and Biotechnology, Budapest University of Technology and Economics, 1521 Budapest, Hungary; ssamaniegoandrade@edu.bme.hu; 2Institute of Materials and Environmental Chemistry, Research Centre for Natural Sciences, Magyar Tudósok Körútja 2, 1117 Budapest, Hungarybakos.istvan@ttk.hu (I.B.); klebert.szilvia@ttk.hu (S.K.); kun.robert@ttk.hu (R.K.); mohai.miklos@ttk.hu (M.M.); 3Department of Chemical and Environmental Process Engineering, Faculty of Chemical Technology and Biotechnology, Budapest University of Technology and Economics, 1521 Budapest, Hungary; 4H-Ion Research, Development and Innovation Ltd., Konkoly-Thege út 29-33, 1121 Budapest, Hungary

**Keywords:** biomass, porous texture, heteroatoms, graphene derivative, surface chemistry, electrical conductivity, AAEMFC, oxygen reduction reaction, lithium-ion battery

## Abstract

In this work, we study the influence of reduced graphene oxide (rGO) on the morphology and chemistry of highly porous N,S-doped carbon cryogels. Simultaneously, we propose an easily upscalable route to prepare such carbons by adding graphene oxide (GO) in as-received suspended form to the aqueous solution of the ι-carrageenan and urea precursors. First, 1.25–5 wt% GO was incorporated into the dual-doped polymer matrix. The CO_2_, CO, and H_2_O emitted during the thermal treatments resulted in the multifaceted modification of the textural and chemical properties of the porous carbon. This facilitated the formation of micropores through self-activation and resulted in a substantial increase in the apparent surface area (up to 1780 m^2^/g) and pore volume (up to 1.72 cm^3^/g). However, adding 5 wt% GO led to overactivation. The incorporated rGO has an ordering effect on the carbon matrix. The evolving oxidative species influence the surface chemistry in a complex way, but sufficient N and S atoms (ca. 4 and >1 at%, respectively) were preserved in addition to the large number of developing defects. Despite the complexity of the textural and chemical changes, rGO increased the electrical conductivity monotonically. In alkaline oxygen reduction reaction (ORR) tests, the sample with 1.25 wt% GO exhibited a 4e^−^ mechanism and reasonable stability, but a higher rGO content gradually compromised the performance of the electrodes. The sample containing 5 wt% GO was the most sensitive under oxidative conditions, but after stabilization it exhibited the highest gravimetric capacitance. In Li-ion battery tests, the coulombic efficiency of all the samples was consistently above 98%, indicating the high potential of these carbons for efficient Li-ion insertion and reinsertion during the charge–discharge process, thereby providing a promising alternative for graphite-based anodes. The cell from the 1.25 wt% GO sample showed an initial discharge capacity of 313 mAh/g, 95.1% capacity retention, and 99.3% coulombic efficiency after 50 charge–discharge cycles.

## 1. Introduction

The fascinating microstructural and surface properties of biomass-based carbon materials justify the extensive interest in their wide range of application in various fuel cells [1,2,3], supercapacitors [4,5], and alkaline, zinc, or ammonium ion batteries [6,7,8,9,10,11,12,13,14]. Compared to terrestrial biomass the potential of marine-based sources, despite their abundance, has been far less explored. Crustaceans rich in N-containing chitin or edible red seaweed with reasonable sulfated polysaccharide content also may serve as precursors for N- or S-containing porous carbon materials [15,16,17].

Porous carbons with a high surface area, rich porosity, and tuned surface chemistry are essential to boost their electrochemical properties. The high surface area, which goes hand in hand with a microporous texture and facilitates the exposure of active sites, might, however, limit the diffusion-controlled processes. In contrast, macroporous materials exhibit better transport kinetics, but their correspondingly lower surface area hampers the number of exposed active sites and therefore results in poorer surface area related performance [18]. Well-tailored porous carbons with interconnected pore morphology make it possible to fabricate high-performance electrodes in fuel cells, such as battery anodes [19] and high-rate electrochemical capacitive energy storage devices [20,21]. In fuel cells the ultra-micro- and micropores are pseudo-active sites for the adsorption and cleavage of O_2_ molecules, while the meso- and macropores promote the oxygen reduction reaction (ORR) by facilitating the mass transfer [22,23,24]. Ultra-micropores of similar size to O_2_ molecules act as pseudo catalytic centers. They stimulate the O_2_ adsorption and consequently promote O=O bond splitting, fostering the ORR through the 4 e^-^ reduction mechanism [25]. The enhanced interface between the electrolyte and the electrodes shortens the ion transport pathways, fosters electrolyte penetration, and facilitates mass transport and ion diffusion in Li-ion batteries as well as in supercapacitors [26,27,28,29]. Excessive widening of the pores decreases the coulombic efficiency and thus the reversible capacity of batteries; however, pores that are too narrow restrict access to the bulky solvated Li-ions. Fine tuning of the carbon microstructure is therefore necessary for an effective balance between the rate performance and storage capacity [30]. The bulky Li-ions solvated in a tetrahedral arrangement within the electrolyte [31] become intercalated and stored in the nanopores of the electrode in de-solvated form. The maximum Li-ion storage capacity can be achieved when the actual size of the Li-ion fits perfectly into the pore size. The benefits of carbon aerogels as anode materials in lithium-ion batteries can be manifested in three aspects: (i) the carbon skeleton is a good electronic conductor that can ensure efficient electron transport; (ii) the 3D interconnected and open porous framework facilitates the electrolyte penetration and provides fast Li-ion diffusion channels; (iii) the open interconnected pores offer enough space to accommodate the volume changes (expansion and contraction during repeated charge–discharge processes) of the electroactive materials, thus helping to maintain the structural and thereby electrochemical stability during cell cycling [32,33,34].

Aside from the pore structure, the other key factor is the surface chemistry of the carbons. The fundamentals of both texture and surface chemistry design and the corresponding challenges of nanostructured mesoporous carbon materials including carbon gels and their application were summarized by Enterría et al. [35]. While the parameters of the sol–gel synthesis technique can be used to tune the adsorption and diffusion characteristics of carbon aerogels, non-metallic heteroatoms (e.g., B, N, O, P, S) generate additional active sites and defects, increase the activity and/or selectivity of the electrochemical processes, and construct more channels for electron and ion transfer [36,37,38].

Although the role of the various N species in the electrocatalytic oxygen reduction process is not fully elucidated, the graphitic and/or pyridinic nitrogen atoms seem to be the most efficient in the ORR [39,40,41,42]. Pyridinic nitrogens present in the carbon structure are Lewis bases, while graphitic or quaternary N may act as n-type dopants in amorphous carbon structures. The higher electronegativity of N (3.04 vs. 2.55 for C) facilitates electron transfer and increases the charge density on the neighboring carbon atom, thus fostering the electrical conductivity, polarity, and surface wettability. In consequence, N doping enhances the high-rate capability in Li-ion battery anode applications and the pseudo-capacitance in supercapacitor applications [43,44,45,46].

The large covalent radius of the S atoms alters the electronic and metallic properties of the carbon matrix [47]. Sulfoxide, sulfones, and sulfonic acids located in the mesopores foster the access of the electrolyte delivering dissolved oxygen into the pore system [48]. The thiophenic sulfur compounds appear to be particularly active in enhancing the physisorption of the oxygen from electrolytes in small carbon pores [48]. The S atom incorporated in aromatic rings causes a slightly positive charge on the neighboring C atoms [48,49] and brings additional hydrophobicity to the surface, thus promoting the adsorption of molecular oxygen [42].

DFT calculations revealed that dual S and N doping resulted in a large number of active sites through the redistribution of spin and charge densities, thus synergistically improving the ORR performance [49]. The atomic level distribution of (hetero)atoms in the porous carbon matrix significantly improved the rate of the ORR, which was otherwise too slow for practical applications [50,51]. They also affected the performance of supercapacitors [52]. They reduce or may even prevent the need for Pt loading, which is an economic obstacle in commercializing fuel cells. N, P, and S atoms in biomass-based carbon electrodes act as extra active Li-ion storage sites [53,54]. The kinetics of the electrochemical reaction in heteroatom doped carbons are much faster, resulting in a higher capacity and better rate performance. The C-S defect sites induce torsion in the graphitic layer structure, and the extended interlayer spacing facilitates the insertion of lithium ions during the charge–discharge cycling of the cells [55].

Nanocarbons (carbon nanotubes, carbide, and carbonitride graphene) themselves may form an electrically conductive interconnected aerogel matrix through van der Waals interactions [56,57]. These particles can be also used as templating agents [58]. Graphene, due to its outstanding properties (high electrical and thermal conductivity and good mechanical strength) plays a particularly important role in the development of a new generation of batteries. Their surface area, mechanical flexibility, and broad electrochemical window [59,60,61] make graphene-based electrodes promising for fuel cells [62], batteries [63,64,65], supercapacitors [66,67,68,69], etc. However, in polar media, graphene tends to aggregate [70]. Its hydrophilic derivative, graphene oxide (GO), owing to its oxygen-containing functional groups, is easier to disperse in the precursors. The thermal/annealing treatment during the synthesis converts the GO to reduced GO (rGO), and thus, the desired structure and physicochemical properties of the electrocatalytically more attracting graphene are at least partly restored [71]:(1)$$GO\to rGO+CO_{2}+CO+H_{2}O.$$

Incorporation of 0.23–0.46 wt% well-dispersed GO into the resorcinol–formaldehyde (RF) polymer xerogel precursor resulted in a carbon xerogel of high porosity and of excellent electrical conductivity: when used as an electrode in aqueous supercapacitors at high current density, the capacitance and the power were enhanced by 25 and 100%, respectively, compared to the undoped carbon xerogel [72]. It was also concluded that the electrical conductivity seemed to play a more important role than the specific surface area in the electrode performance of mesoporous carbon xerogels. GO loadings exceeding 2 wt% resulted in lower energy and power densities in spite of the increased electric conductivity [73]. In Li-ion batteries, graphene-doped carbon/carbon aerogel electrodes can accommodate lithium more easily than the common graphite anode [59], as in addition to the intercalation mechanism, they exhibit fast lithium adsorption [31,74,75,76], defect trapping [75], build-up of faradaic capacitance [77,78], etc.

In spite of the numerous studies on the performance of electrodes fabricated from carbons doped either with two heteroatoms or with carbon nanoparticles, their simultaneous application is rare [79,80], and none of the studies addressed the influence of the nanoparticle concentration.

Earlier we observed that while incorporation of N or reduced GO alone in a resorcinol–formaldehyde-based carbon aerogel did not alter the 2e^−^ the mechanism of the ORR, when both of them were present, the dominant pathway changed from the slow 2e^−^ to the more efficient 4e^−^ route [81]. Recently, we found that a carbon electrode fabricated from an ι-carrageenan–urea biopolymer hydrogel precursor displayed almost the same oxygen reduction reaction (ORR) performance as a commercial 20 wt % Pt/C electrode. The electrode was found to be stable and exhibited a 4-electron behavior in alkaline ORR [82].

Based on these findings, in this work, we focus on the influence of rGO on the morphology and chemistry of highly porous carbon materials obtained from the same biopolymer cryogel. Simultaneously, we propose an easily upscalable route to prepare rGO doped N,S containing biopolymer based carbons. In reference works, the incorporated carbon nanoparticle content was studied in a wide range [72,73,79,80]. Previously, we found that in responsive 3D poly(N-isopropylacrylamide) networks the percolation threshold was around 5 wt% [83]. Therefore, in this work the GO content was varied between 1.25 and 5 wt%. The texture of the carbon cryogels was characterized by electron microscopy imaging, N_2_ and CO_2_ adsorption measurements, Raman spectroscopy, and powder X-ray diffraction (XRD). X-ray photoelectron spectroscopy (XPS), SEM-supported electron dispersive spectroscopy (EDS), ultimate elemental analysis, and FTIR were used to study the surface and bulk chemistry. The samples were probed as electrodes (i) in a fuel cell under condition, similar to alkaline anion exchange membrane fuel cells (AAEMFCs) and (ii) in Li-ion batteries.

## 2. Materials and Methods

### 2.1. Synthesis

rGO containing N,S double-doped carbon aerogels were synthesized, using ι-carrageenan (commercial grade, Sigma Aldrich, Budapest, Hungary), urea pearls (98%, Sigma Aldrich, Budapest, Hungary), and aqueous GO suspension with a method adapted from Li et al. [17]. A detailed scheme of the synthesis procedure is presented (Figure 1).

The stock aqueous GO suspension (0.96 wt%) was prepared from natural graphite (Graphite Týn, Týn nad Vltavou, Czech Republic, 95%) using an improved Hummers’ method [84,85]. Hydrogels were obtained by mixing 2 g urea and 2 g ι-carrageenan with 100 mL aqueous GO suspension (containing 50, 100, and 200 mg GO, respectively) at 80 °C. A GO-free gel was also prepared for comparison. The cooled hydrogels with 1.25, 2.5, and 5 wt% GO content were freeze-dried and pyrolyzed in a rotary quartz reactor at 700 °C (20 °C/min) in dry N_2_ flow (25 L/h) for 1 h. After washing with aq. 1.0 M HCl, the samples were annealed at 1000 °C in Ar flow for 1 h [82]. The resulting carbon cryogels are labeled as CA, CAGO50, CAGO100, and CAGO200.

### 2.2. Characterization Methods

Scanning electron micrographs were taken by a Zeiss Sigma 300 FESEM field emission scanning electron microscope (Carl Zeiss QEC GmbH, Oberkochen, Germany). Low-temperature (−196.15 °C) nitrogen adsorption measurements were performed after 24 h degassing at 110 °C on a NOVA 2000e (Quantachrome, Boynton Beach, FL, USA) automatic volumetric instrument. The apparent surface area *S_BET_* was determined using the Brunauer–Emmett–Teller (BET) model [86]. The pore volume *V_0.98_* was estimated from the amount of vapor adsorbed at *p/p_0_* = 0.98, assuming that the adsorbed gas fills the pores as liquid. The Dubinin–Radushkevich (DR) plot [87] was used to calculate the micropore volume *V_micro_*. The pore size distribution was computed by the Quenched Solid Density Functional Theory (QSDFT) for slit/cylindrical pore geometry [88]. Carbon dioxide adsorption was measured at 0 °C up to atmospheric pressure with an AUTOSORB-1 (Quantachrome, Boynton Beach, FL, USA) computer-controlled analyzer. Evaluation of the primary adsorption data was performed with the Quantachrome AsiQwin software (version 3.0). Raman spectra were obtained using a LabRAM (Horiba Jobin Yvon, Kyoto, Japan) instrument. The laser source was a λ = 532 nm Nd-YAG (laser power at the focal point 15 mW). A 0.6 OD filter was used to reduce the power of the beam. Parameter optimization and data analysis were performed by LabSpec 5 software. Powder X-ray diffractograms (XRD) were obtained in the range 2θ = 10°–130° with an X’Pert Pro MPD (PANalytical Bv., Almelo, The Netherlands) X-ray diffractometer using an X’celerator type detector and monochromatic Cu Kα radiation with a Ni filter foil (λ = 1.5406 Å).

The ultimate elemental analysis was performed in triplicate on an Elementar Vario Micro (CHNS) instrument. The lack of metal traces was confirmed with SEM/EDS (JEOL JSM 6380LA, Tokyo, Japan). X-ray photoelectron spectra were recorded on a Kratos XSAM 800 spectrometer operating in fixed analyzer transmission mode, using Mg K_α1,2_ (1253.6 eV) excitation. The pressure of the analysis chamber was lower than 1 × 10^−7^ Pa. Survey spectra were recorded in the 150–1300 eV range in 0.5 eV steps. The photoelectron lines of C1s, O1s, N1s, and S2p were measured in 0.1 eV steps with 1 s dwell time. The spectra were referenced to the energy of the C1s line of the sp^2^ type graphitic carbon, set at 284.3 ± 0.1 eV binding energy (BE). Peak decomposition was performed after Shirley-type background removal using a Gaussian–Lorentzian peak shape with a 70:30 ratio. Details of the applied fitting procedure are described elsewhere [89]. Quantitative analysis, based on integrated peak intensity, was performed by the XPS MultiQuant program [90], applying the conventional infinitely thick layer model using the experimentally determined photoionization cross-section data of Evans et al. [91] and the asymmetry parameters of Reilman et al. [92]. Attenuated total reflectance Fourier transform infrared spectroscopy (FTIR-ATR) were recorded on powdered carbons in the 4000–400 cm^−1^ wavenumber range at a resolution of 4 cm^−1^ by 32 scans using a Tensor 27 (Bruker Optik GmbH, Leipzig, Germany) spectrophotometer equipped with a Platinum ATR unit A225. The crystal was made of diamond having a refractive index of 2.4. For the background signal, the measured medium was air. Since absorption by the powders was very strong, a moderate polynomial baseline correction and smoothing were required.

### 2.3. Electrochemical Performance Tests

The electrical conductivity was estimated by a laboratory made instrument. The powdered samples were gradually compressed (0.5–5 MPa) in a rigid polytetrafluoroethylene (PTFE) tube (1 cm^2^ internal cross section) with two copper bolts (one on each side of the tube). The conductivity was estimated from the pressure dependence of the resistance [93]. The electrocatalytic ORR tests were performed using a glassy carbon (GC) rotating disc electrode (RDE, Pine Research Instrumentation, Durham, NC, USA). The ink for the working electrodes was prepared by dispersing 2 mg powdered annealed carbon (CA, CAGO50, CAGO100, or CAGO200) in a mixture of 1.6 mL MilliQ water, 0.4 mL isopropyl alcohol, and 8 µL 5% Nafion^®^ solution. After 30 min sonication, the ink was pipetted onto the dry-mirror-polished GC and dried at room temperature. The loading varied between 50 and 400 μg/cm^2^. The measurements were implemented in 0.1 M KOH electrolyte using three-electrode systems with a hydrogen electrode as a reference and Pt wire as a counter electrode in a three-compartment PTFE cell [82]. All potentials are given vs. the reversible hydrogen electrode (RHE).

The working electrodes for the Li-ion storage studies were prepared by mixing the carbon samples and polyvinylidene fluoride binder (PVDF 99.9%, Solvay, Budapest, Hungary) in a weight ratio of 95:5 in a ball mill for 1 h. After adding an adequate amount of N-methyl-2-pyrrolidone (NMP 99.9%, Sigma-Aldrich, Budapest, Hungary), the slurry was cast on to a copper foil using an automatic film applicator (BYK Gardner GmbH, Geretsried, Germany). The coated film was dried overnight at 70 °C in a vacuum oven. Then, 14 mm diameter circular discs were cut from the dried film using a manual disc cutter (Berg &Schmidt GmbH, Hamburg, Germany). CR2032 coin-type cells were assembled in an argon-filled glove box using the as-prepared coated disc and Li metal disc (99.9%, Sigma-Aldrich, Budapest, Hungary) as working and counter electrodes, respectively. Whatman glass fiber served as the separator, and 1.0 M lithium hexafluorophosphate (LiPF_6,_ 99.95%, Sigma-Aldrich, Budapest, Hungary) in ethylene carbonate (EC, 98%, Sigma-Aldrich, Budapest, Hungary) and diethyl carbonate (DEC, 99%, Sigma-Aldrich, Budapest, Hungary) in 1:1 v% served as the electrolyte. The cells were assembled using a manual crimping machine (MTI MSK 110, Xiang’an, China). The electrochemical performance of the half-cell was performed using an electrochemical workstation (Biologic VMP 300, Seyssinet-Pariset, France) at room temperature. The galvanostatic charge–discharge cycling of the half-cell was conducted in a potential window of 0–3 V (vs. Li^+^/Li).

## 3. Results and Discussion

### 3.1. Development of the Morphology during the Annealing

The FESEM images demonstrate the complexity of the pore structure of the annealed carbon gels (Figure 2). The early inclusion of GO already in the precursor suspension influences the morphology developing during gelation and cryogenic drying and is reflected also in the final annealed state (Figure 2c). It is apparent that the reduced GO formed during the heat treatments was evenly distributed in the carbon matrix. The typical nanopores within the CA matrix were 2–3 nm (Appendix A). The yield of the synthesis steps is reported in Appendix A.

The low temperature (−196 °C) N_2_ adsorption isotherms of the annealed carbon samples having various GO content are shown in Figure 3a. All the isotherms are of composite Type IV + Type II with an H4 hysteresis loop displaying a sharp step at *p/p*_0_ = 0.45 [94]. The shape and type of the hysteresis loops suggest an interconnected pore network of micro-, meso-, and macropores. Numerical data deduced from the measured isotherms are presented in Table 1. Since the macropores are not totally filled with condensed nitrogen, the liquid equivalent volume *V_0.98_* was determined at *p/p_0_* = 0.98.

The most spectacular effect of the GO was in the micropore region of the isotherm, which ran almost parallel in the mesoporous region. Incorporation of GO increased not only the carbon content of the polymer matrix but also the volatile and gaseous products that were released during the thermal treatments (see also Equation (1)). The CO, CO_2_, and water act as self-activating agents during their path through the material and alter both its texture and chemistry. The 1.25–2.5 wt% GO samples (CAGO50 and CAGO100) increased the pore volumes and the apparent surface area almost proportionally, while addition of 5 wt% GO (CAGO200) resulted in a drop in both the surface area and pore volume. In this case, the amount of evolving oxidative reactants was too high and overactivated the sample [70]. Additionally, the ultra-micropore region was revealed by the CO_2_ adsorption measurements performed at 0 °C. The combined pore size distribution curves in Figure 3b indicate the presence of ultra-micropores as well as significant micro- and narrow mesoporosity in the range of 1.7–6 nm. The increasing amount of oxidative volatile gases evolving during the thermal decomposition of the GO may also be responsible for the decreasing trend in ultra-microporosity.

The Raman spectra in Appendix A exhibit the iconic D (~1350 cm^−1^; defects, edges, and disordered carbon sites) and G (~1580 cm^−1^; vibration of sp^2^-hybridized graphitic carbon) band regions typical of carbon materials [95]. The first and second order regions of the spectra were deconvoluted for further analysis (Appendix A [96,97,98,99]). The D band (around 1350 cm^−1^) indicates the aromatic clusters with edges/borders of mainly amorphous carbon (sp^3^-bonded) structures [100]. The blue shift of the G band may be caused by internal stress developing when the incorporated GO and/or newly developing clusters collide during the annealing treatment at 1000 °C [100,101]. The D′ band, associated with disordered graphenic lattices (around 1620 cm^−1^) appeared only in the GO-free CA sample. On the other hand, the second-order region was detected only in the rGO-containing samples. The trend in the *I*_D_/*I*_G_ ratio along with the relative sharpening of the G band and the reduced intensity D2033 in the GO-added samples confirm an ordering effect of the forming rGO in the carbon matrix [9].

### 3.2. Surface Chemistry of the Samples

The effect of the GO on the surface composition was studied by XPS (Table 2). The successful removal of the metal impurities was confirmed by SEM-supported ED spectroscopy (Appendix A). As expected, the O content of the incorporated GO affected not only the porous texture but also the surface chemistry of the annealed samples. The complexity of the thermal degradation and high temperature chemical reactions are responsible for the change in the overall C or O content and the trend of the heteroatom/carbon ratios.

The composite photoelectron lines (C1s, O1s, N1s, and S2p) were decomposed into different chemical states. The self-oxidation altered the surface composition as well as the chemical environment. Due to the complexity of the samples, however, the chemical states could not be resolved exactly. The suggested states referred to a broad range of chemical bonds. The shape of the C1 (sp^2^ C) component was asymmetric, as in pure and low heteroatom-containing graphite and graphene. The tail of the asymmetric peak overlapped with the peaks of the C–O(N) components; using this line-shape also gave a satisfactory oxygen + nitrogen balance (the ratio of the measured and calculated concentrations) for all samples.

Figure 4 presents a typical decomposition scheme of the C1s, O1s, N1s, and S2p spectral regions, while Table 3 and Table 4 show the quantitative results of the decomposition, together with the binding energy ranges of the components and their assignations to various chemical states.

The decomposition of the C1s signal revealed three different carbon species. The most abundant was the sp^2^ form (C1). The presence of the π-π^*^ shake-up satellite also confirmed the assignation of this component. The concentration of C1 was slightly higher in the rGO-containing samples. The other components could be assigned to carbon atoms connected to one (C2) or two (C3) heteroatoms.

The O1 and O2 oxygen components were found in all samples; they could be assigned to the S–O and to various C–O bonds, respectively. The O3 component, present in the high rGO-containing samples, could be connected to either highly oxidized carbon (anhydride or carbonate). Owing to their low concentration, the corresponding carbon species could not be resolved in the spectra. Nitrogen was also present in three chemical states. N1 was the simple C–N bond, while N2 was bonded to carbon with further heteroatoms. The quantities of these nitrogens were commensurable. The low-concentration high-binding energy N3 component was a kind of quaternary ammonium. Due to the synthesis procedure, it is assumed that the nitrogen atoms were homogeneously distributed in the matrix and also decorated the walls of the micropores and ultra-micropores [102]. The sulfur content, present as C–S sulfide (S1) and in oxidized forms (S2; sulfone, sulfate), was the least affected by the treatments, but it should be noted that all the sulfur in the CAGO50 sample was sulfidic, while in the others, the contribution of the S2 forms reached 20–30%.

The FTIR measurements give complementary information about the chemical structures of the samples with diverse GO content (Appendix A, Table 4). From the rich spectra, three main regions were considered to monitor the chemical changes in the bulk [103,104]. All spectra displayed a strong adsorption band at 1554 cm^−1^, assigned to the –C=C– skeletal vibration of aromatic regions. The 1749 cm^−1^ band of the GO-loaded samples implied the presence of carbonyl groups (–C=O), stemming from aldehydes, ketones, esters, and carboxylic acids. The band at 1372 cm^−1^, which was absent in the starting material, was ascribed to the –OH in-plane bending of phenols. If we compare the amount of the latter two signals to the corresponding carbon skeleton peak at 1554 cm^−1^, we can conclude that the ratio of oxygen-containing groups increases with the increasing GO content.

In summary, addition of GO to the ι-carrageenan–urea carbon xerogel resulted in multifaceted modification of the textural and chemical properties of the samples. The thermal decomposition of GO enhanced self-activation and facilitated the formation of micropores which, for the CAGO50 and CAGO100 samples, also resulted in a considerable increase in the apparent surface area. With 5 wt% GO, too much oxygen was probably introduced into the matrix which destroyed the finer porosity. The annealing at 1000 °C removed part of the heteroatoms but left behind a high number of defects. It is expected that these new active sites (N-S-C) “constructed” from edged thiophene S, graphitic N, and pentagon defects, as well as the 3D porous architecture, result in an enhanced electrocatalytic activity [17].

### 3.3. Effect of GO on ORR Performance

Like other groups, we also found that the incorporation of rGO improved the electric conductivity of the cryogels [73,105]. In our samples, the improvement was proportional to the amount of GO added, in spite of the lower apparent surface area of CAGO200 (Appendix A).

The ORR performance of the samples was investigated under conditions similar to the alkaline anion exchange membrane fuel cells (AAEMFCs). The linear sweep voltammograms (LSVs) of the four samples are plotted in Figure 5. The high potential regions revealed not only the slight differences between the various samples but also their alteration during the consecutive cycles. CAGO200 was found to be particularly unstable. The addition of GO gradually shifted the onset potential. CAGO50 exhibited the highest half-wave potential, which gradually decreased back to the value observed with the GO-free CA carbon. In Table 5, we compare the onset and half-wave potentials (*E*_1/2_) of these curves with the results of other groups.

The Koutecky–Levich (KL) equation [114] was used to estimate the correlation between the current density *j* and the rotation rate ω
(2)$$\frac{1}{j}=\frac{1}{j_{k}}+\frac{1}{j_{lim}}=\frac{1}{j_{k}}+\frac{1}{0.62\times n\times F\times C{\times D}^{\frac{2}{3}}\times {\nu^{-\frac{1}{6}}\times \omega }^{\frac{1}{2}}},$$
where *j*_k_ is the kinetic current density, *j*_lim_ is the limiting diffusion current density, *n* is the number of electrons transferred in ORR per oxygen molecule, F is the Faraday constant, *D* is the diffusion coefficient of oxygen in the electrolyte, *ν* is the kinematic viscosity of the electrolyte, and *C* is the concentration of oxygen in the electrolyte. The KL plots of CAGO50 (Figure 6) displayed a slope closest to the theoretical curve corresponding to the 4e^−^ mechanism, i.e., the incorporation of 1.25% GO improved the performance, whereas further GO had the opposite impact. At higher GO loadings, the lines became steeper, nearer to the theoretical 2e^−^ slope. Based on these findings from our samples, CAGO50 seems to have the optimum pore texture and chemistry for ORR under the investigated conditions. Although the electrical conductivity increased linearly with the increasing GO loading, no similar change in ORR was apparent. This agrees well with the work by Ramos-Fernández et al., who also observed that increasing GO content improved the electric conductivity of carbon xerogels, but enhanced porosity may spoil the electrochemical performance by disrupting the rGO network throughout the carbon matrix [73].

The (in)stability of the electrodes was further explored with CV responses measured after a series of treatments simulating application conditions. Based on the KL evaluation the GO-free CA and the CAGO50 and CAGO200, showing the lowest and highest slopes were selected for stability studies. The fresh electrodes made from the carbons, after running a CV cycle (*curve 1*) were exposed to further 50 cycles in argon-treated oxygen-free KOH electrolyte and then to another 50 cycles in oxygen-saturated KOH. Finally, an ORR test was run (sweep rate: 5 mV/s, rotation rates: 400, 625, 900, 1250 rpm). The voltammograms, *curves 2, 3,* and *4*, obtained after each set of treatments are compared in Figure 7. (The effect of the electrode loading is discussed in the Appendix A) The slightly distorted rectangular shapes may stem from the pseudo-faradaic reactions of the corresponding functional groups [115]. The width of the hysteresis is not correlated with the BET surface area (Table 1), but the presence of the GO apparently reduces the gravimetric capacitance. The coincidence of *curves 1* and *2* in CA carbon (Figure 7) revealed that this sample was stable under oxygen-free conditions. After an additional 50 cycles in oxygen-saturated electrolyte (*curve 3*), the widening of the CV curve implies that the electrocatalytically active surface area increased in the presence of oxygen. The consecutive four ORR cycles further modified the electrode surface (*curve 4*).

The performance of the CAGO50 carbon was very similar, but it seemed somehow even less stable under oxygenic conditions. The CAGO200 electrode was unstable already in oxygen-free conditions (*curves 1* and *2*). The change in the surface was even more pronounced after being cycled in the oxygen-saturated electrolyte, which apparently stabilized the surface: no further changes were seen after these ORR cycles (*curve 4*). These observations indicate that the electron transfer mechanism deduced from the KL plot can only be considered as an approximation. The kinetics of the reduction reaction in the presence of the heteroatoms is indeed more sophisticated and further influenced by the rGO. The chemical species locally evolving in the ORR may initiate various processes including restructuring, activation, degradation, etc., of the electrode surface. The plots in Figure 7 clearly show that even if the electrode was stabilized under oxygen-free conditions, it might later undergo destabilization processes during the ORR. Nevertheless, the widening of the CV curves marked an increase in the electrocatalytically active surface area. The deviation of the corresponding curves confirmed that the addition of GO to the precursor affected the stability of the electrodes. As expected from the KL plot, the CAGO200 sample was more sensitive to the oxidative conditions, but after stabilization this sample exhibited the highest capacitance. The trend of the gravimetric capacitance in this figure was different from the trend expected from the apparent surface area (1070, 1479, and 933 m^2^/g for CA, CAGO50, and CAGO200, respectively), implying that the accessibility of the nitrogen molecules is different from that of the species involved in the electrocatalytic processes [116].

### 3.4. Effect of the GO on Li-Ion Battery Application

In order to reveal the effect of GO on the performance of the carbon cryogel as anode material in Li-ion batteries, galvanostatic charge–discharge cycling tests were performed. In the half-cell setup, the CA, CAGO50, CAGO100, or CAGO200 sample was the working electrode, and the Li metal disc served as the counter and reference electrode, i.e., no further conductive agent was used. In the galvanostatic test, a constant current *I* is applied to the cell and the potential is evaluated as a function of time *t*. The specific capacity *C_s_* during complete charge–discharge is given by
(3)$$C_{s}=\frac{I\times t}{m}.$$

The nature of the charge–discharge plots largely depends on the surface chemistry (microstructure, porosity, heteroatom content, etc.) and the adsorption capacity of the carbon host. Figure 8 shows the cell voltage vs. the specific capacity charge–discharge cycling plots of the samples under investigation. While highly ordered graphitized carbon anodes show a steep charge–discharge curve [117], sloping shapes similar to Figure 8 are distinctive for amorphous biomass-derived carbon anodes in Li-ion battery applications [33,118,119]. The broad peaks in the XRD patterns (Appendix A) reveal that the carbon cryogels studied in this work were typically amorphous; therefore, charge–discharge curves with this shape were expected, i.e., the CA, CAGO50, CAGO100, and CAGO200 have the potential to store the Li-ions gradually and consistently over their operating potential. A comparable charge–discharge plateau was measured also on RF polymer-based carbon xerogels [120]. The latter microporous carbon xerogels were successfully tested as anode material in lithium-ion cells exhibiting a specific capacity 288 mAh/g and retaining 97% of the initial capacity after 100 charge–discharge cycles.

The cyclability of the carbons is presented here as the total charge–discharge capacity (or capacity retention) as a function of the number of cycles (Figure 9). The undoped carbon, CA showed an initial discharge capacity of 332 mAh/g at a constant current density 100 mA/g. The cell showed good capacity retention and high coulombic efficiency (92.9% and 99.7%, respectively) even after 50 charge–discharge cycles.

For comparison, 218 mAh/g at a current density 0.372 A/g was reported by Wang et al. [6]. Among the GO doped carbons, CAGO50 exhibited the best performance in terms of the cycling stability. The cell showed an initial discharge capacity of 313 mAh/g and a highly reversible capacity of 297 mAh/g after 50 cycles, corresponding to a 95.1% capacity retention. This good stability agreed well with the impedance analysis results: this carbon had the lowest impedance. The enhanced performance of CAGO50 compared to the pristine CA can be explained in terms of the surface area and porosity [121]. The higher surface area of CAGO50 improved the electrolyte accessibility and ion diffusion in the pores of the electrode, which in turn resulted in a higher coulombic efficiency and cell stability [30,121]. The charge–discharge plateau recorded in the first and 50th cycle was almost identical, confirming the resilient stability of the electrode [122]. Following this, the CAGO200 showed an initial discharge capacity of 240 mAh/g and a capacity retention of 93.6% after 50 cycles (Figure 9).

The overall electrochemical performance of CAGO200 in the Li-ion battery tests was better than the undoped CA and slightly lower than the CAGO50. The CAGO100 presented the poorest electrochemical performance in spite of having the largest surface area. This apparent surface area, however, was also combined with a larger pore volume in the wider pore size region, which seemed to decrease the initial coulombic efficiency and reduce the reversible capacity of the battery [30]. The initial discharge capacity of CAGO100 was 255 mAh/g, but the cell maintained its stability poorly. The low retention capacity (67.2% after 50 cycles) might be due to a kinetic limitation in the porous electrode [121]. The compatibility of the pore and the Li^+^ ion sizes is also responsible for the charge capacity. CAGO100 and CAGO200 showed the highest and lowest pore volume, respectively, both in the micro- and mesopore regions. On the one hand, wider pores may not be completely filled by the Li^+^ ions, and on the other, low porosity limits the entrance of Li^+^ containing electrolyte into the pores. CAGO50, which seems to have the optimum pore size compatible with Li^+^ and is capable of effectively accommodating the Li^+^ ions, exhibited the highest overall capacity among the GO-doped samples.

As shown in Figure 9, the coulombic efficiency of all the samples was consistently maintained above 98.5% indicating the efficient Li-ion insertion and reinsertion potential of these carbons during the charge–discharge process [122]. CA and CAGO100 displayed lower coulombic efficiency than CAGO50 and CAGO200 (>99.3%) indicating that the addition of GO was clearly beneficial for more stable electrochemical performance in Li-ion batteries, but the GO doping level must be carefully tuned.

It may be concluded that the carbons investigated here can be utilized as potential anodes in Li-ion batteries. According to our results, the pore structure and the chemical composition corresponding to CAGO50 proved to be the best to achieve an acceptably high reversible discharge capacity and electrode stability under prolonged cycling. The electrochemical performance of these carbon electrodes is comparable to the previously reported results on carbon electrode anodes in Li-ion batteries [123,124,125,126,127].

## 4. Conclusions

Addition of GO to the precursor biopolymer successfully increased the surface area and the porosity of the carbon material up to 2.5 wt% GO (up to *S*_BET_ 1780 m^2^/g and *V*_0.98_ 1.7 cm^3^/g), but the high amount of gases released when 5 wt% GO was added destroyed the finer pores. The oxidative species evolving from the GO during the heat treatments influenced the surface chemistry in a complex way, but sufficient O, N, and S heteroatoms were preserved in various chemical forms. The highest concentration of sp^2^ carbons was detected on the surface of sample CAGO50 (1.25 wt% added GO). Although the volatile thermal degradation products of the GO affected the porous texture and the surface chemistry in a subtle way, the residual rGO resulted in a monotonically increasing electrical conductivity, in spite of the notable drop in the surface area. The rGO affected the electron transfer mechanism and compromised the stability of the electrodes in ORR tests. The CAGO50 sample displayed the advantageous 4e^-^ mechanism. The CAGO200 (5 wt% GO) sample was the most sensitive under oxidative conditions, but after conditioning, it exhibited the highest capacitance in ORR. In the Li-ion battery tests, the coulombic efficiency of all the samples was consistently above 98%. The cell made from CAGO50 showed an initial discharge capacity of 313 mAh/g, 95.1% capacity retention, and 99.3% coulombic efficiency after 50 charge–discharge cycles. Despite having the largest surface area sample, the CAGO100 performed more poorly than the CAGO50 in both application tests. This is a clear indication that concerted fine-tuning of the pore morphology and surface chemistry is required for both advanced electrocatalytic and electrochemical performance. The as-prepared GO-doped carbon samples with their ultra-micropores and high surface area performed well and could be promising electrode materials for both applications.

## Figures and Tables

**Figure 1 nanomaterials-13-02364-f001:**
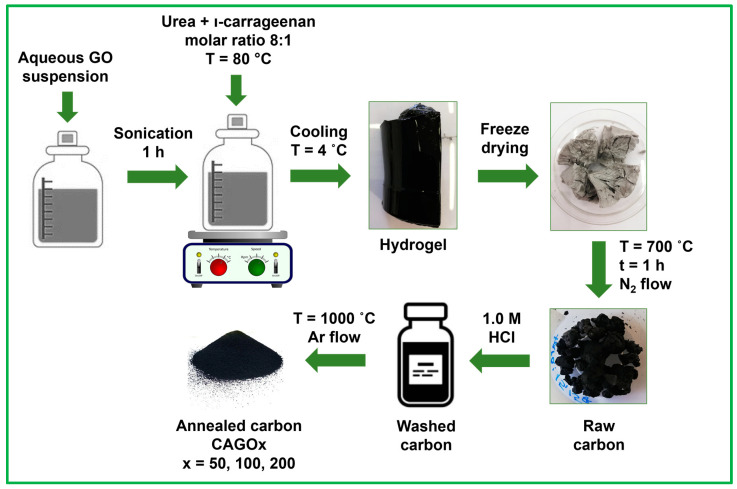
Scheme of the synthesis procedure. The GO is added in as-received suspended form, and the biopolymer is formed in aqueous medium, making the synthesis route green and easily upscalable.

**Figure 2 nanomaterials-13-02364-f002:**
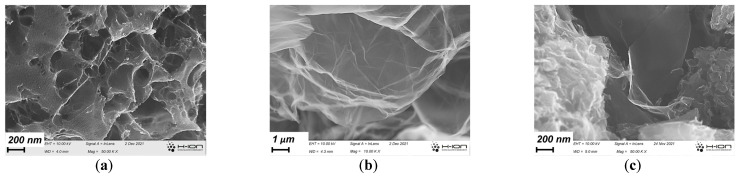
Typical images of CA (**a**), the GO cryogel (**b**), and the CAGO50 sample (**c**).

**Figure 3 nanomaterials-13-02364-f003:**
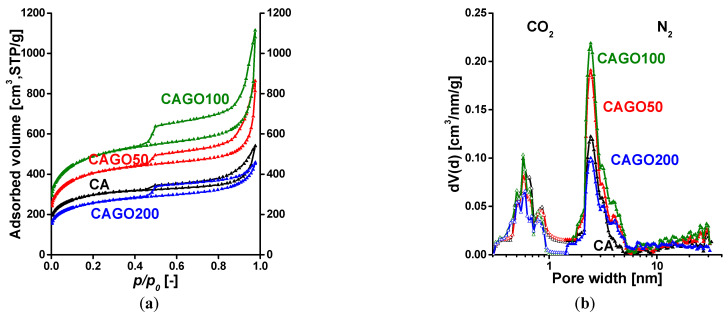
(**a**) The low temperature N_2_ adsorption/desorption isotherms of annealed carbons; (**b**) the combined pore size distribution functions of annealed carbons. Functions were estimated by quenched solid density functional theory QSDFT, slit/cylinder geometry for N_2_, and NLDFT for CO_2_, respectively.

**Figure 4 nanomaterials-13-02364-f004:**
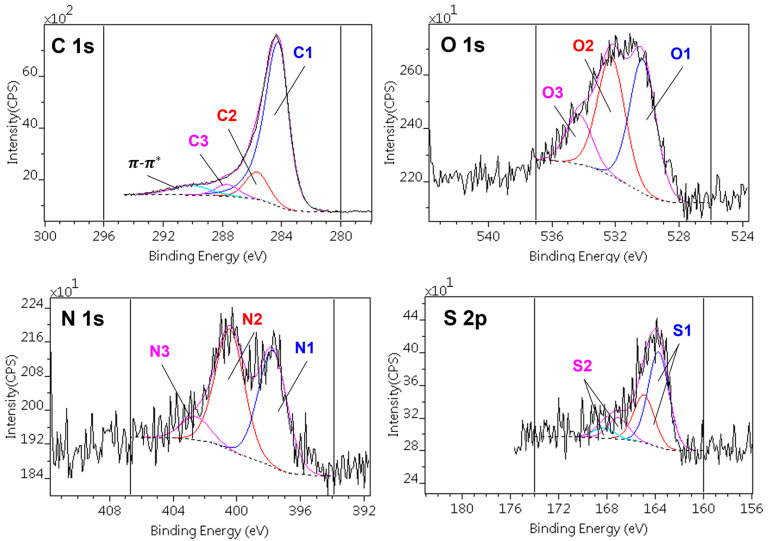
Decomposition of the C1s, O1s, N1s, and S2p regions of the photoelectron spectra of the CAGO100 sample.

**Figure 5 nanomaterials-13-02364-f005:**
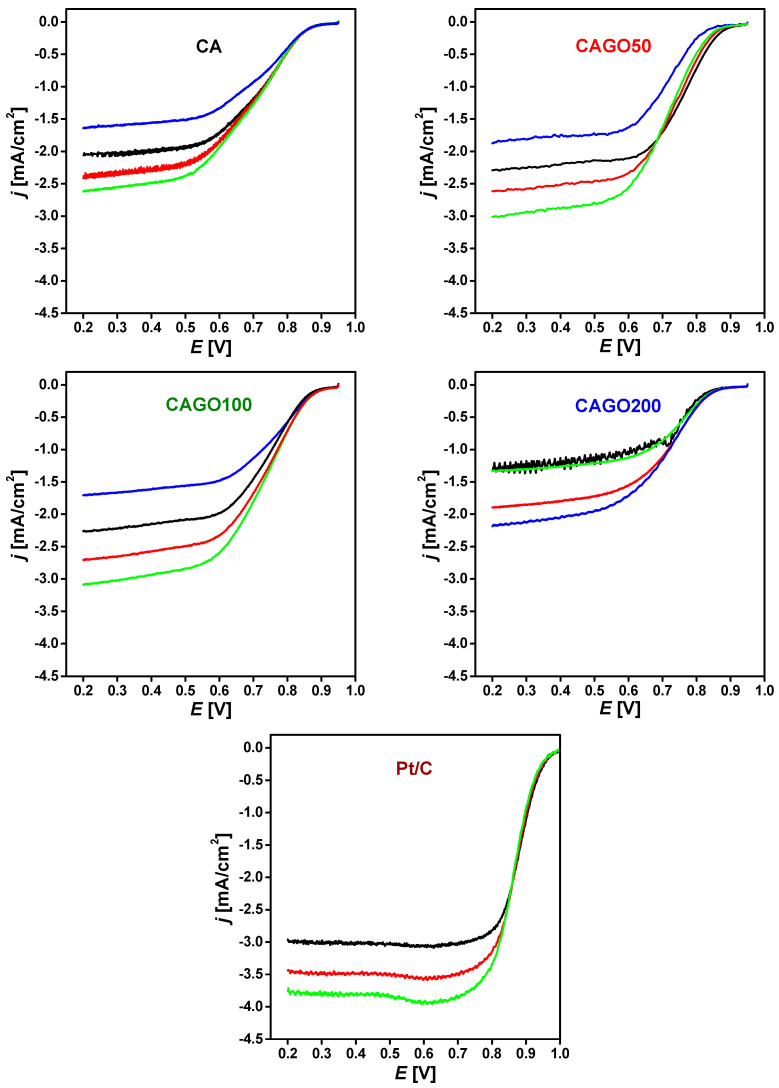
Cathodic linear potential sweep of the carbon xerogel loaded electrodes in oxygen saturated 0.1 M KOH. Loading: 75 μg/cm^2^, sweep rate: 5 mV/s. Rotation rate: 400 (blue), 625 (black), 900 (red), 1250 (green) rpm, (increasing downwards). Rotation rates were applied in the following order: 625, 900, 1250, and 400 rpm. The LSVs on Pt/C were measured for comparison.

**Figure 6 nanomaterials-13-02364-f006:**
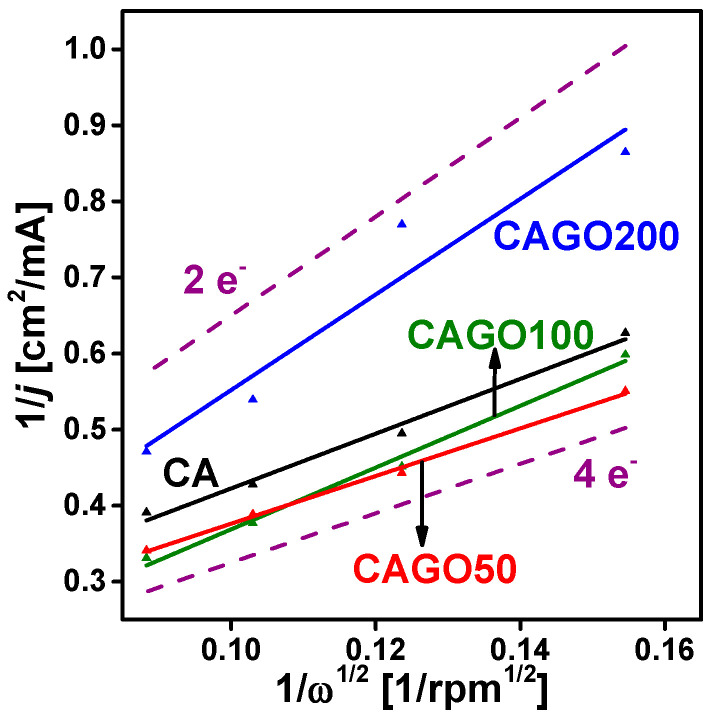
Koutecky–Levich (KL) plots of the carbon electrodes at *E* = 0.3 V. The theoretical 2e^−^ and 4e^−^ KL plots are shown for comparison.

**Figure 7 nanomaterials-13-02364-f007:**
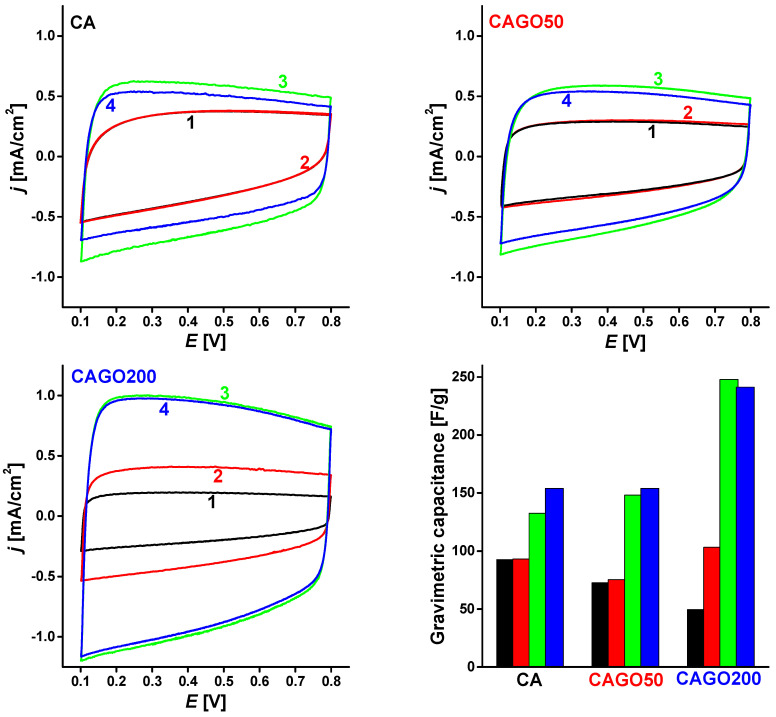
Cyclic voltammograms (1) of the virgin electrode; (2) after 50 cycles in oxygen-free 0.1 M KOH electrolyte; (3) after an additional 50 cycles in oxygen-saturated electrolytes; (4) after an additional ORR test with rotating electrode (sweep rate: 5 mV/s, four rotation rates). The column graph compares the gravimetric capacitances of the carbons corresponding to CV 1 (black), 2 (red), 3 (green), and 4 (blue), respectively.

**Figure 8 nanomaterials-13-02364-f008:**
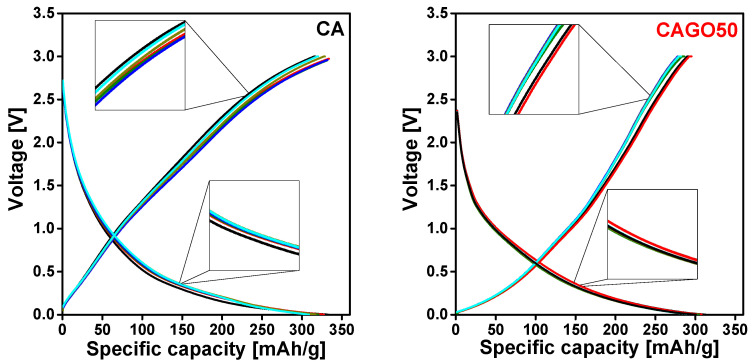
Galvanostatic charge–discharge profiles of CA, CAGO50, CAGO100, and CAGO200 at a constant current density of 100 mA/g, cycle 1 black, cycle 10 red, cycle 20 olive, cycle 30 blue, cycle 40 dark yellow, and cycle 50 cyan.

**Figure 9 nanomaterials-13-02364-f009:**
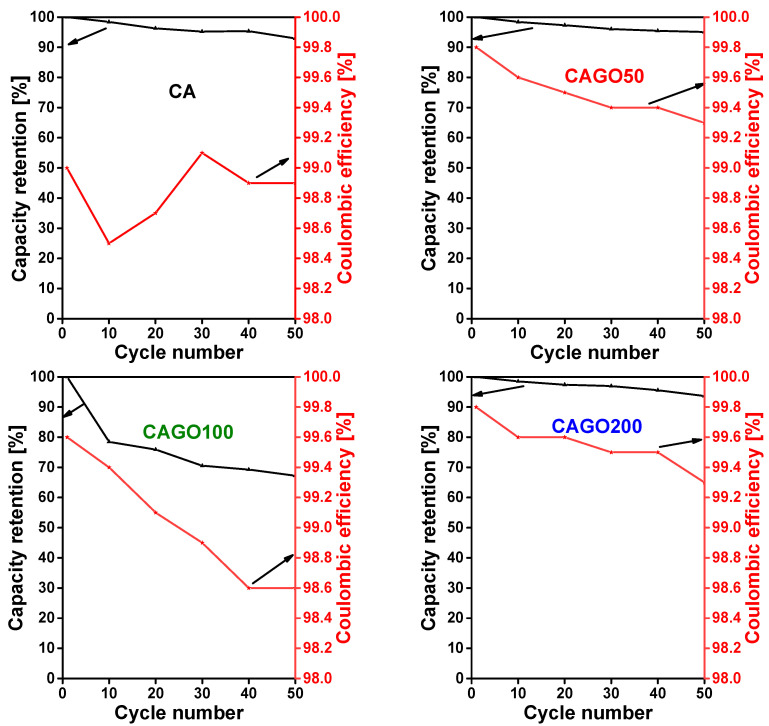
Long-term cycling performance of CA, CAGO50, CAGO100, and CAGO200 at a constant current density of 100 mA/g. Capacity retention black, Coulombic efficiency red.

**Table 1 nanomaterials-13-02364-t001:** Porous characteristics of annealed carbon aerogel samples from gas adsorption measurements ***.

Method	Parameter	Units	CA	CAGO50	CAGO100	CAGO200
**From N_2_**	*S_BET_*	[m^2^/g]	1070	1479	1779	933
	*V_0.98_*	[cm^3^/g]	0.83	1.33	1.72	0.71
	*V_micro,DR_*	[cm^3^/g]	0.42	0.54	0.64	0.34
		[%]	51	40	37	48
	*V_micro,DFT_*	[cm^3^/g]	0.31	0.40	0.48	0.25
		[%]	37	30	28	35
**From CO_2_**	*V_umicro,DR_*	[cm^3^/g]	0.073	0.062	0.059	0.041
	*V_umicro,DFT_*	[cm^3^/g]	0.042	0.039	0.030	0.025

* *S_BET_:* apparent surface area from the BET model, *V_0.98_*: liquid equivalent of the gas adsorbed at *p/p_0_* = 0.98, *V_micro,DR_*: micropore volume from the DR model, *V_micro,DFT_*: micropore volume from the DFT model; *V_umicro,DR_*: ultra-micropore volume from the DR model; *V_umicro,DFT_*: ultra-micropore volume from the DFT model.

**Table 2 nanomaterials-13-02364-t002:** Surface composition (atomic %) measured by XPS.

Sample	C	O	N	S	O/C	N/C	S/C	O + N + S	S/N
								C	
CA	90.6	3.3	5.1	1.0	0.036	0.056	0.011	0.104	0.196
CAGO50	92.0	3.1	3.7	1.3	0.034	0.039	0.014	0.087	0.361
CAGO100	90.7	4.1	4.1	1.2	0.045	0.045	0.013	0.104	0.293
CAGO200	90.4	3.7	4.4	1.4	0.041	0.049	0.015	0.105	0.318
GO film	67.4	32.1	-	0.5	0.476	-	0.007	0.484	-

**Table 3 nanomaterials-13-02364-t003:** Decomposition of the C1s and O1s regions of the photoelectron spectra: binding energy ranges, chemical state assignations, and surface compositions (atomic %).

	C1s			O1s		
	C1	C2	C3	O1	O2	O3
**Chemical state**	sp^2^ C=C	C–O C–N C–S	C=O O–C–O N–C–O	S–O	C–O–C C–OH C=O	OC–O–CO (H_2_O)
**Binding energy [eV]**	284.3–284.4	285.7–285.8	287.5–287.9	530.2–530.6	532.1–532.5	533.9–534.3
CA	74.0	10.9	5.4	1.5	1.7	-
CAGO50	78.8	7.4	5.5	1.9	1.3	-
CAGO100	74.7	11.0	4.8	1.8	1.7	0.7
CAGO200	75.9	9.4	4.8	1.8	1.6	0.5

**Table 4 nanomaterials-13-02364-t004:** Decomposition of the N1s and S2p regions of the photoelectron spectra: binding energy ranges, chemical state assignations, and surface compositions (atomic %).

	N1s			S2p	
	N1	N2	N3	S1	S2
**Chemical state**	C–N	OO–C–N	C–N^+^	C–S	C–SO_3_
**Binding** **energy [eV]**	397.8–398.0	400.4–400.5	402.4–402.7	164.9–165.0	168.3–168.6
CA	2.3	2.3	0.8	0.9	0.2
CAGO50	1.6	1.7	0.6	1.2	-
CAGO100	1.9	1.9	0.4	1.0	0.2
CAGO200	2.0	2.0	0.7	1.1	0.3

**Table 5 nanomaterials-13-02364-t005:** Comparison of the onset potentials and halfwave potentials of double-doped carbon aero/xero/cryogel electrodes in 0.1 M KOH, vs. RHE.

Sample	BET Surface Area [m^2^/g]	Onset Potential [mV]	*E*_1/2_ [mV]	Number of e^−^ Transferred	Ref.
SWCNT@N,P doped carbon	616	920	850	3.91	[79]
N,S co-doped 3D rGO	392	895	732	3.87	[80]
N,S porous carbon materials	732	940	840	-	[106]
N,P-holey graphene foams	758	983	865	3.70	[107]
3D-high performance graphene	1406	928	836	3.83	[108]
N,P porous graphitic biocarbon	845	−14 (vs. Ag/AgCl)	−115 (vs. Ag/AgCl)	3.9	[109]
N,P co-doped carbon	375	950	820	3.7	[110]
N,P,S co-doped carbon nanosheets	1198	938	800	3.8–4.0	[111]
P,N,S-porous carbon	711	905	780	3.68–3.96	[112]
N,P,B biocarbon	1155	904	790	3.78–3.90	[113]
Pt/C (20 wt.% Pt on Vulcan XC-72)	-	960	869	3.96	[113]
CA	1070	855	700	3.5	Our work
CAGO50	1479	850	760	4.0	
CAGO100	1779	845	730	3.1	
CAGO200	933	825	700	2.0	
Pt/C		957	886		

## Data Availability

The data presented in this study are available on request.

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
