# Peer review of "The Influence of Reduced Graphene Oxide on the Texture and Chemistry of N,S-Doped Porous Carbon. Implications for Electrocatalytic and Energy Storage Applications"

_nanomaterials, 2023, doi:10.3390/nano13162364_

Round 1
Reviewer 1 Report
The authors synthesized porous carbon doped with graphene oxide using the same substrate, and studied the effect of different content of graphene oxide on the morphology and properties of the material. After carefully reading, it was found that this article needs major revision because several issues and explanations are still need to be clarified. I recommend it publication in this journal (MAJOR REVISION) after providing proper improvement in revised version by including suggestion, modification and reply to raised queries which are given below.
1. The title of the paper is too cumbersome and needs to be changed to reflect the theme and highlights of the paper.
2. The abstract section needs to be more succinct in expression, describing the impact of key variables on the results and looking ahead to the implications of the study. In addition, the expression of some sentences needs to be revised, such as "challenged the stability of the electrodes".
3. The introduction should include a brief description of the research background, the problems encountered, a comparative analysis of the existing schemes, the proposed scheme, and an introduction to the advantages, feasibility and significance of the proposed scheme.
4. Authors need to distinguish RGO and GO.
5. The purity of raw materials used in the experiment should be provided, and the manufacturer, model and parameters should be indicated for special reagents and instruments.
6. The specification of Figure 1 and the font color of the ruler need to be changed. Please refer to the requirements of the journal for the specific style.
7. The author needs a clearer analysis and explanation for the addition of 5% GO (CAGO200) resulted in a drop both in surface area and pore volume.
8. It is recommended to use a three-line table and avoid complete duplication with the data in the picture.
9. Please modify the font and format of Figure 3, 4 and other pictures, and reformat according to the requirements of the journal. The overall effect of the combined picture should be considered, and the resolution of the combined picture should be at least 600dpi.
10. For Figure 7, the line color is difficult to distinguish, and it is recommended to adjust to a more contrasting color.
11. Please pay attention to the writing of subscripts and superscripts.
12. The authors should be careful with the Hyphen, minus, en-dash, and em-dash in the whole manuscript.
13. There are some grammatical errors in this manuscript. The English language should be improved.
14. Some of the references are not close related to the theme of this manuscript and suggest to be replaced by close related ones, e.g. Journal of Bioresources and Bioproducts 2022, 7 (4), 245-269; Rare Metals 2022, 41 (10), 3432-3445; Coordination Chemistry Reviews 2022, 466, 214604.
Minor editing of English language is required.
Author Response
Please see the attachemnt

Reviewer 2 Report
In this manuscript, aiming the energy storage, authors prepared high surface area carbon doping with GO. The influence of GO was studied in ORR and Li ion storage applications. Comprehensive characterizations have been performed. In general, it is an interesting work and the manuscript is well prepared. However, there are still some issues to be addressed. A moderate revision is suggested before its acceptance.
1. One or two sentences are required at the beginning of abstract to present the background or aim of this work.
2. One or two more keywords can be added.
3. One separate paragraph can be added to briefly introduce the novelty, strategy, method and important results.
4. The generally introduction of the different energy storage sources should be provided with some more recent supporting articles, such as aqueous Zn-ion batteries (Journal of Alloys and Compounds, 2022, 903: 163824); lithium–selenium batteries (Rare Metals, 2022, 41(10): 3432-3445); Li-ion battery (New Journal of Chemistry 45, 19446-19455, 2021); Zn-air battery (Molecules 28 (5), 2147, 2023); supercapacitor (Journal of Bioresources and Bioproducts, 7, 4, 245-269, 2022); ammonium-ion battery (Chemical Engineering Journal, 2023, 458, 141381); etc.
5. One scheme to show the experimental procedure is suggested for better understanding of this work to readers.
6. Three-line tables should be applied for a better scientific expression.
7. Why authors used GO in this work should be further clarified with the structure, properties, modifications and applications of GO with some recent supporting articles: Molecules 28 (6), 2535, 2023; European Polymer Journal 141 (5), 110083, 2020; Molecules 27 (24), 8896, 2022; etc.
8. The gap areas and the new contribution in the paper should be further clarified.
9. There are too many too old references, which is better to be deleted or replaced with recent articles to show the novelty of this work.
10. There are still some typos and grammar issues in the manuscript. Authors should carefully recheck the whole manuscript.
Round 2
Reviewer 2 Report
Authors have address most of the issues. However, the following issues should be further modified.
1. Two more advanced energy storage sources should be provided in introduction, such as aqueous Zn-ion batteries (Journal of Alloys and Compounds, 2022, 903: 163824); Li-ion battery (New Journal of Chemistry 45, 19446-19455, 2021); etc.
2. All the references should be written with all authors' names, such as ref. 10. Please recheck the similar issue all the reference list.
